

Trends in evapotranspiration and its drivers in Great Britain: 1961 to 2015
Eleanor M. Blyth, Alberto Martinez-de la Torre, Emma L. Robinson
Centre for Ecology and Hydrology, Maclean Building, Benson Lane, Crowmarsh Gifford, Wallingford,
OX10 8BB, UK
*Correspondence to: Eleanor M. Blyth (emb@ceh.ac.uk)*



Abstract
In a warming climate, the water budget of the land is subject to varying forces such as increasing
evaporative demand, mainly through the increased temperature, and changes to the precipitation,
which might go up or down.
Using a verified, physically based model with 55 years of observation-based meteorological forcing,
an analysis of the water budget demonstrates that Great Britain is getting warmer and wetter.
Increases in precipitation ($3.0\pm2.0$ mm yr$^{-1}$ yr$^{-1}$) and air temperature ($0.20\pm0.13$ K decade$^{-1}$) are
driving increases in river flow (2.16 mm yr$^{-1}$ yr$^{-1}$) and evapotranspiration (0.87 mm yr$^{-1}$ yr$^{-1}$), with no
significant trend in the soil moisture.
The change in evapotranspiration is roughly constant across the regions whereas runoff varies
greatly between regions: the biggest change is seen in Scotland (4.56 mm yr$^{-1}$ yr$^{-1}$), where
precipitation increases were also the greatest ($5.4\pm3.0$ mm yr$^{-1}$ yr$^{-1}$) and smallest trend (0.29 mm yr$^{-1}$
yr$^{-1}$) is seen in the English Lowlands (East Anglia and Midlands), where the increase in rainfall is not
statistically significant ($1.1\pm0.7$ mm yr$^{-1}$ yr$^{-1}$).
Relative to their contribution to the evapotranspiration budget, the increase in interception is higher
than the other components. This is due to the fact that it correlates strongly with precipitation which
is seeing a greater increase than the potential evapotranspiration. This leads to a higher increase in
actual evapotranspiration that the potential evapotranspiration, and a negligible increase in soil
moisture or groundwater store.



## 1. Introduction
Evapotranspiration affects many important physical aspects of the land: the dryness of the soil,
vegetation growth, the temperature of the air and the amount of water in the rivers and
groundwater reserves. It is therefore important to understand how evapotranspiration responds to
changing climate.
For Great Britain (hereafter GB), studies based on the National River Water Archive have shown an
overall increase in riverflow of 1.6 mm $yr^{-1}$ $yr^{-1}$ over the last five decades (Hannaford, 2015).
Meanwhile, an analysis of the meteorological data for the years 1961-2012 has shown an overall
increase in potential evaporation for GB over that time of 0.77±0.77 mm $yr^{-1}$ $yr^{-1}$ (Robinson et al,
2017a). With an increase of rainfall at 2.86±0.65 mm $yr^{-1}$ $yr^{-1}$ (Robinson et al, 2017a) and assuming
the evapotranspiration increases roughly in line with potential evaporation (Kay et al, 2013), this
would imply an increase in soil moisture or groundwater recharge of about 0.5 mm $yr^{-1}$ $yr^{-1}$.
There is uncertainty in two of these estimates: firstly, the riverflow record is based on a sub-set of
rivers and the remainder – particularly in the Scottish Highlands which contribute a large portion of
the water – is gap-filled using a simple model. Secondly, the actual evapotranspiration may not
follow the potential evaporation. The estimate made by Kay et al (2013) was using a simple model
with little representation of vegetation processes.
To answer the question about how the actual evapotranspiration has changed over the last 55 years,
we would ideally have a direct observation of it both at the long term and over a representative
large scale. However, evapotranspiration is difficult to measure. Direct observations of evaporation
in GB from flux-tower data exist over short time periods or over small areas, but there are no large-
scale, long term observations of this elusive flux.
Since there are few long term observations of evapotranspiration, the only way to study its evolution
and response to changes in climate is by using a model.
However, evapotranspiration is also difficult to model as it depends on the trio of soil, vegetation
and atmospheric conditions and the interactions between them. Key processes include the
availability of soil moisture to plants during dry spells, the response of plants to temperature,
sunshine and soil moisture, the intercepted of rainfall by plants. Assumptions about the processes
involved and their interactions (see Wang and Dickinson, 2012 for a review of methods) can have a
significant impact on the resulting modelled evaporation (see Schellekens et al, 2017 for overview of
model differences).





A previous study using a simple soil-moisture-stress-based model (Kay et al, 2013) indicated that
there has been an increase in evaporation over the last few decades. However, this modelled trend
is only due to changes in soil moisture stress and evaporative demand. The overall trend might be
altered by other aspects of the vegetation-atmospheric interactions such as rainfall interception and
transpiration responses to changes in meteorology and soil moisture stress.
In this paper, a comprehensive land surface model will be used to diagnose the changes in
evapotranspiration in GB over 55 years (1961 to 2015) including the analysis of the different
components of evapotranspiration (bare soil, transpiration, interception).
The questions that will be addressed are as follows:
1.   Is the evaporation of GB and the regions increasing or decreasing?
2.   Which components of the evaporation are contributing to the trend?
3.   What meteorological changes are driving these changes?
In Sect. 2, the model, the ancillary and the driving data will be described before the model outputs
are presented which will be evaluated with available data. In Sect. 3, the resulting trends in the
model outputs will be analysed to address the three questions formulated above. Sect. 4 then
discusses the implications of these results while Sect. 5 presents the conclusions about the study.





2 Method
This study uses a physically-based model (JULES: Joint UK Land Environment Simulator, Best et al,
2011, Clark et al, 2011) driven by observation-based driving data for 55 years (see Sect. 2.1). The
accuracy of the model will be assessed by comparing to a range of independent datasets (see Sect.
2.2 for mean monthly evapotranspiration fluxes based on flux-tower data from 4 contrasting sites,
annual regional runoff-data from river-flow data, and Sect. 2.3 for estimates from other large-scale
models).
2.1 JULES model description and setup
The JULES model includes many of the processes that are likely to affect changes in water loss (see
Appendix A). It is used here in a new assessment of the GB water balance for the years 1961 to 2015.
Figure 1 show maps of the time-average quantities of evapotranspiration, sensible heat, soil
moisture and soil temperature and Figure 2 shows the spatial average annual values of runoff,
evapotranspiration, soil moisture and soil temperature. The rest of the paper describes the
provenance of this figure, and analyses the implications of the results. The model output, ancillary
files and meteorological forcing data are collectively referred to as CHESS (Climate, Hydrological and
Ecological research Support System).



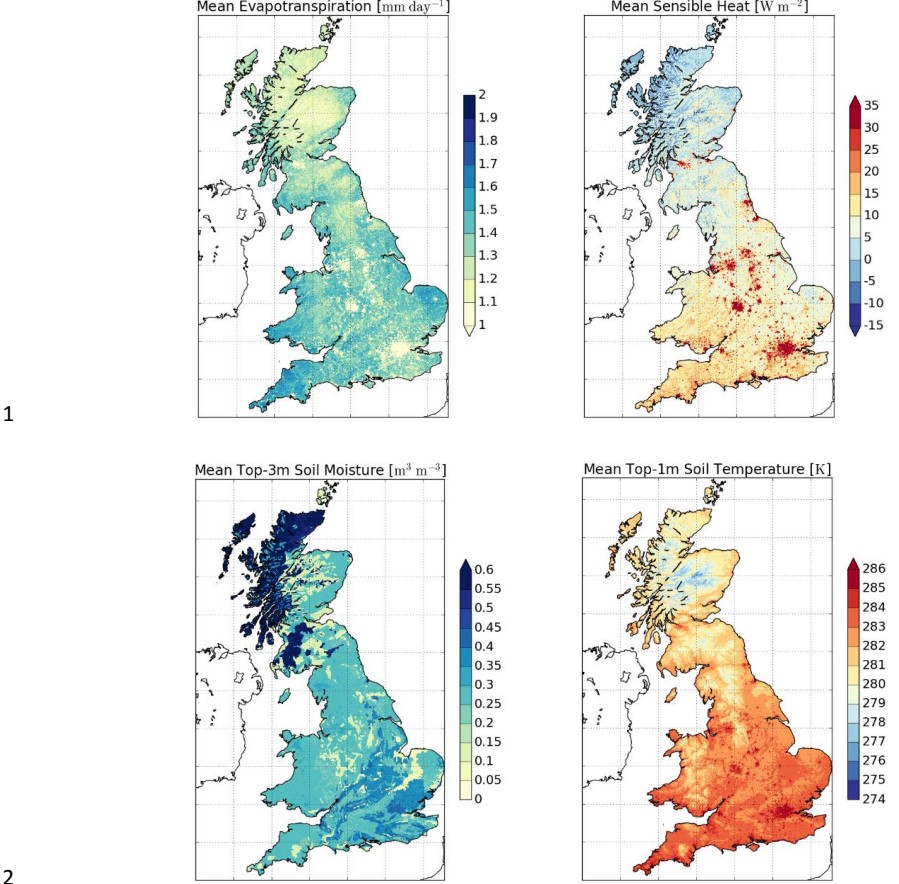

Figure 1. Top and middle rows: Modelled estimates of mean evapotranspiration, sensible heat, soil moisture and soil temperature averaged over 1961 to 2015 for GB.

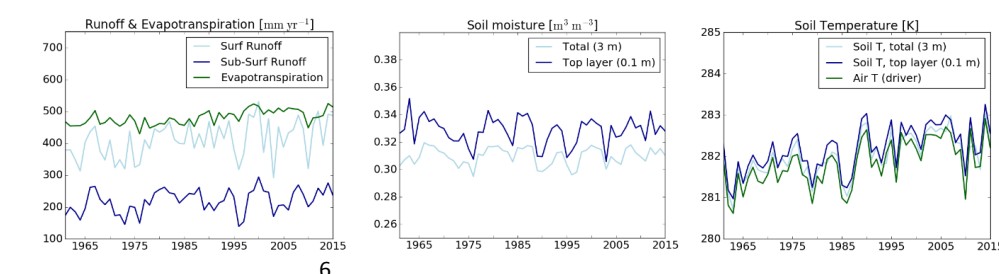

Figure 2. Modelled estimates of runoff, evapotranspiration, soil moisture and soil temperature, averaged over GB against year.



Using land cover maps (CEH Land Cover 2000: Fuller et al, 2002), soils maps (Harmonised World Soil
Database: FAO/IIASA/ISRIC/ISS-CAS/JRC, 2012) and a newly disaggregated meteorological driving
data (Robinson et al, 2017a), the model has been run at a landscape scale of 1 km. This spatial scale
is a compromise between being small enough to make reasonable comparisons with observational
datasets (such as flux towers), and large enough to be able to obtain reasonable meteorological
driving data and with a feasible computer cost.
The model used a fixed vegetation map based on observations in the year 2000 (CEH Land Cover
2000: Fuller et al., 2002) to prescribe the land cover fractions of the 8 different categories: broad
leaf trees, needle leaf trees, grass, crops, shrub, water, bare soil and urban. Parameters values for
these land covers are part of the model configuration (see Appendix B).
The phenology for each month was prescribed for the deciduous vegetation and the crops. The soil
hydrology component of JULES is based on the Darcy Richards equations (see Appendix A for a
summary) with the vertical discretization into four layers. The parameters used in the equations
depend on the soil type and maps of these were derived from the Harmonised World Soil Database
(see Appendix B for a description of their provenance).
The meteorological dataset used in the simulations is described in Robinson et al (2017a). This
dataset is a combination of daily precipitation based on observations (Gridded Estimates of daily and
monthly Areal Rainfall: GEAR: Keller et al, 2006) and other meteorological data from the
observation-based product MORECS (Thompson et al, 1981, Hough and Jones, 1997). The MORECS
data is presented at 40km and for CHESS, the data is downscaled using information about
topography. Robinson et al (2017a) analysed the CHESS data and showed a positive trend in the air
temperature as well as a positive trend in short wave radiation, which is due to increasing short
wave radiation in the spring months. The short wave radiation is based on sunshine hours and the
CHESS data includes a spatially varied impact of aerosol loading that affects the relationship
between sunshine hours and short wave radiation. However, it does not explicitly include any time-
variation of the aerosols (dimming and brightening). To some extent the dimming and brightening is
included implicitly as the sunshine hours are above a certain threshold of intensity, and therefore
overall lower light levels will result in fewer hours. This is discussed in more detail in Sect. 4. The
data have since been extended to 2015 (Robinson et al, 2017b).
The model represents a significant upgrade to the current product available which employed a much
simpler water balance model: the Met Office Regional Evaporation Calculation Scheme (hereafter
MORECS: Thompson et al, 1981, Hough and Jones, 1997), that does not represent photosynthetic



processes, has a simple soil-physics routine and runs on a 40 km grid square on a daily time step. The
MORECS evapotranspiration is used widely and provides an interesting point of comparison for this
new evapotranspiration data product (see Sect. 2.3).
2.2 Evaluating the model results with observations
This paper makes no attempt to calibrate the model further than has already been done for global
simulations. But it is instructive for this analysis to quantify the performance of the model with a
standard configuration. Ideally we would be able to evaluate the model with observations at the
1km scale with daily or monthly data. However, it is only possible to make direct observations of
evapotranspiration using eddy-correlation systems which observe the hourly fluxes over an area of
about 100m. At the other extreme, river-flow data can be used in combination with the precipitation
to imply the evapotranspiration of a catchment (~100 km) over a year. In this section, the results of
the modelled actual evapotranspiration are compared to data from four eddy-correlation systems
and the river-flow records of the UK.
2.2.1 Evaporation from eddy-correlation systems.
There are several flux sites across GB. Out of these, four have been selected, based on the following
criteria: a good energy closure, running for several years when the CHESS data is available and
represents contrasting land cover types (trees and grass) and regional variation (Scotland and
England). Their locations are shown in Fig. 3, and Table 1 lists the data that are available, the dates
and location coordinates of each site.

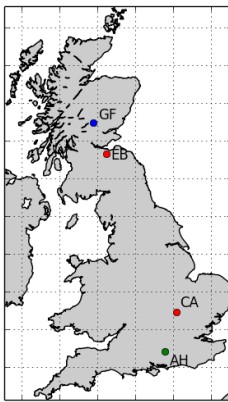

Figure 3: Map showing the positions of the flux sites used for validation. AH=Alice Holt,
CA=Cardington, EB=Easter Bush, GF=Griffin Forest. The colours correspond to the vegetation type at
the site: green=broadleaf (BL), deciduous, blue=needleleaf (NL), evergreen, red=grass (GR).





|  | Latitude | Longitude | Years | Land cover at site | % Land cover of CHESS square: BroadLeaf,NeedleLeaf,GRass,Shrub,Crop,Urban |
|---|---|---|---|---|---|
| Alice Holt | 51.1535 | -0.8583 | 1999-2013 | BL | 36*, 2, 24, 0, 38, 0 |
| Cardington | 52.1 | -0.416 | 2005-2011 | GR | 0, 0, 53.5, 7, 25.5, 14 |
| Easter Bush | 55.8660 | -3.2058 | 2003-2005 | GR | 10, 0, 53.5*, 4, 25, 7.5 |
| Griffin Forest | 56.6072 | -3.7981 | 1997-1999 | NL | 0.5, 91*, 0, 8.5, 0, 0 |

Table 1: Sites used to compare with the CHESS data and JULES runs with details of land cover. *
indicates the land cover used in the single dominant vegetation cover runs (Fig. 4).
As each CHESS grid square contains a range of vegetation types, two model results are shown: with
the grid square covered entirely by the vegetation at the site of each flux tower and the grid square
with the fractional vegetation cover as used in CHESS (see Sect. 2.1). The original flux data are
measured at 30 minute intervals; the data are checked for quality and gap-filled before daily
averages are calculated. From the daily averages, mean-monthly values are calculated and
presented here.
Fig. 4 shows the comparison of the model output with the mean monthly observed data.

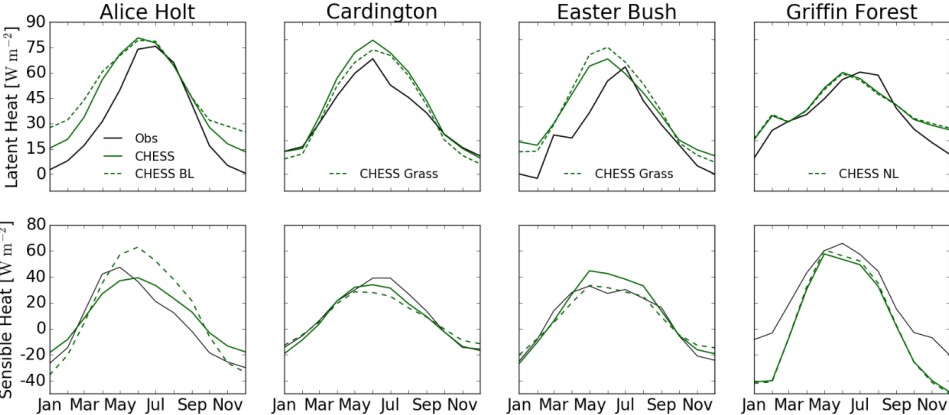



Figure 4: Climatological monthly average latent and sensible heat fluxes for each of the sites (black
solid), compared to the CHESS square (green solid), as well as the same square re-run with the single
dominant vegetation cover (green dashed).
The first thing to note is that CHESS tends to simulate higher evaporation than the observations for
all of the sites. Following Blyth et al (2010) we compare the evaporative fraction (the ratio of
evapotranspiration to the sum of evapotranspiration and sensible heat flux) in Table 2 which allows
for equal underestimation of both sensible and latent heat fluxes. In this case, Griffin Forest and
Cardington overestimate the total evaporation while Alice Holt underestimates it and Easter Bush is
the same. On average however, there is a systematic overestimation of the evaporation in JULES
which has been noted previously (van den Hoof et al, 2013) and is discussed in Sect. 3.
2.2.2 The seasonality of evapotranspiration
Differences in seasonal evapotranspiration between the data and the model are shown in Fig. 4,
which highlight some issues with the model, discussed below.
For the forest sites (Alice Holt and Griffin Forest) the main discrepancy is in the winter when the
model substantially overestimates the evapotranspiration compared to the data: while the
observations indicate values of latent heat around zero to 10 W m$^{-2}$, the model has values of about
20 W m$^{-2}$. In the case of Griffin Forest, the energy for the modelled winter evapotranspiration is
coming from the negative sensible heat flux (around -40 W m$^{-2}$ compared to an observed value of
around -10 W m$^{-2}$). In the case of Alice Holt, the negative winter time sensible heat flux is reasonably
matched by the observations for the run with single dominant vegetation cover (BL; broad leaf
trees). The energy required by the model for the winter evapotranspiration must therefore be due
to an overestimate of the net radiation balance.
In the case of the grass sites (Cardington and Easter Bush), the winter time fluxes are reasonably well
modelled, but the summer modelled evapotranspiration is too high. In Cardington, this
overestimation in latent heat is matched by an underestimation of sensible heat. In Easter Bush,
there is no simultaneous reduction in sensible heat and therefore energy required for the high latent
heat must be due to an overestimate of the net radiation.
The conclusions are the same for both types of model runs (single dominant vegetation cover and
mixed vegetation cover), apart from Alice Holt, where the high winter latent heat fluxes are for the
BL run. This can be explained by the fact that the CHESS square contains substantial fraction of crops
and grasses (62% - see Table 1).



2.2.3 Analysis of modelled components of evapotranspiration at the flux sites
Figure 5 shows the different components of the modelled evapotranspiration (soil surface
evaporation, transpiration and interception). A summary of the results is shown in Table 2.

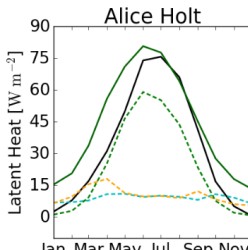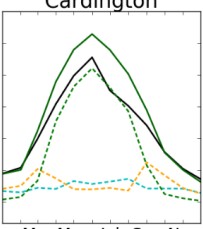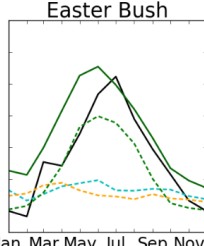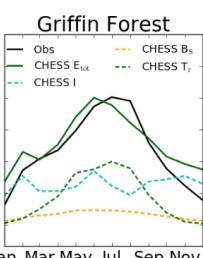

Figure 5: Seasonal variation in evapotranspiration observed (black solid) and modelled (green solid),
and components – interception (cyan dashed), soil surface evaporation (orange dashed) and
transpiration (green solid).

|  | Evaporative Fraction as % | | P (mm yr$^{-1}$) | Transpiration, $T_r$ | | Soil surface evaporation, $B_S$ | | Interception, I | |
|---|---|---|---|---|---|---|---|---|---|
|  | Obs | Model | | % $E_{tot}$ total | % P | % $E_{tot}$ total | % P | % $E_{tot}$ total | % P |
| Alice Holt | 88 | 81 | 832 | 54 | 29 | 24 | 13 | 22 | 11 |
| Cardington | 77 | 85 | 562 | 60 | 45 | 22 | 17 | 18 | 13 |
| Easter Bush | 77 | 77 | 876 | 50 | 21 | 24 | 9 | 27 | 12 |
| Griffin Forest | 61 | 95 | 1215 | 35 | 11 | 13 | 4 | 52 | 17 |

Table 2: Summary of model results at the four flux sites: Annual average evaporative fraction
(modelled and observed) and precipitation (P). Annual average fluxes of modelled transpiration ($T_r$),
soil surface evaporation ($B_S$) and interception (I) as % of total evapotranspiration ($E_{tot}$) and
precipitation (P).
There are no observations of these components at the four flux sites, so it is not possible to evaluate
them directly. However there are estimates of the three components for different vegetation types
in the literature which can be used to assess the model results. In order to translate between the
quoted figures and the model, it is important to note that some authors present the results as the
fraction of the total evaporation and others as the fraction of the total precipitation. For instance,
Van den Hoof et al (2013) summarises work from a range of studies in Europe (Wilson et al, 2001,


Verstraeten et al, 2005, and Choudhury and DiGirolamo, 1998), focusses on the former and quotes
values of forest interception to range from 13% to 25% of the total evaporation while for grasses it is
closer to 10%.
Meanwhile, Nisbet (2005) summarises a large body of work by scientists studying interception and
transpiration in the UK (Calder, 1990, Calder et al, 2003, Roberts, 1983), focusses on the latter and
quotes a percentage of rainfall that is intercepted: about 20% of rainfall for broadleaf trees and 35%
for needleleaf. The evidence suggests that the annual fraction of rainfall intercepted by trees is fairly
constant across a wide range of annual rainfall regimes, although increases slightly at low annual
rainfalls (<500 mm yr$^{-1}$). The data are sparser for grass and no values are given in this report,
although interception rates of heather and bracken are quoted at about 20% of rainfall.
The model results in Table 2 confirm the analysis of Nisbet (2005). The fraction of rainfall that is lost
through interception is fairly constant over the four sites, ranging from 11% to 17%. This translates
into a large fraction of total evaporation that is due to interception: 18% to 52%. At Griffin Forest,
the model evaporative fraction is much higher than the observation (95% compared to 61%). 52% of
the model evaporation is due to interception. Compared to the Van den Hoof et al (2013) figures of
interception, this might be considered to be too high. However, this only represents 17% of the
precipitation in this high-rainfall area. Many of the studies from Van den Hoof are in regions with
much lower annual precipitation. The value of 17% is low compared to other estimates of
interception loss in needle-leaf trees as reported in Nisbet (2005).
In the much lower rainfall regime where Alice Holt is sited, the modelled interception is only 22% of
the evaporation budget, which is dominated by the transpiration (54%).
This analysis suggests that the model has a reasonable representation of interception, capturing its
conservative relationship to the precipitation.
The transpiration, on the other hand, is more controlled by the available energy and has a more
uniform relationship with total evaporation (see Roberts, 1983) ranging from 35% to 60%, and a
wider range of fractions of precipitation, from 11% to 45%. The values of transpiration are lower
than the values given by Van den Hoof et al (2013) and Nisbet (2005) who quote values for trees
from 53% to 70% of total evaporation being due to transpiration. It is possible that the total value of
transpiration is reasonable, but the fraction is low due to the overestimation of the total
evaporation.



The bare soil evaporation compares reasonably well with data quoted in the literature although
figures are rather low.  The values range from 13% to 24% of the total evaporation and correspond
to values quoted in van den Hoof et al (2013) of 20% and 30%. The low bias might be due to the fact
that the UK sites are in relatively high rainfall areas with high interception so that even if the total
evaporation is correct, the percentage is low.
2.2.4 River flow
At the annual timescale, changes in water storage in a UK catchment are negligible and the area-
average evapotranspiration can be estimated as precipitation minus river-flow. Daily river flow
records for the UK are available in the National River Water Archive (NRFA) which is hosted at CEH,
Wallingford. A subset of these records have been chosen for their length of record, accuracy, little or
quantifiably affected by abstractions. The annual runoff from these catchments is then combined
with a model to scale up to the regional and GB scale to present net annual and monthly runoff. The
method for this is described in Marsh et al (2015). Hannaford (2015) has summarised the results,
which are reproduced here in Figs. 6, 8 and Table 3. The challenge addressed here is to identify
whether using a model to fill in the gaps introduces a bias to the results.
Fig. 6 plots the total GB annual river flows from these processed observations. The GB runoff, and
precipitation minus the evapotranspiration (the difference being the soil-moisture) from the model
is also shown. It is apparent from this figure that the modelled runoff is slightly lower than that
observed, which is commensurate with the analysis in the previous sub-section that the modelled
evapotranspiration is too high. However, the percentage difference is very small (0.15%) which is not
consistent with the previous estimates of the bias (roughly 10%). An analysis of the regions shows
where this discrepancy occurs.

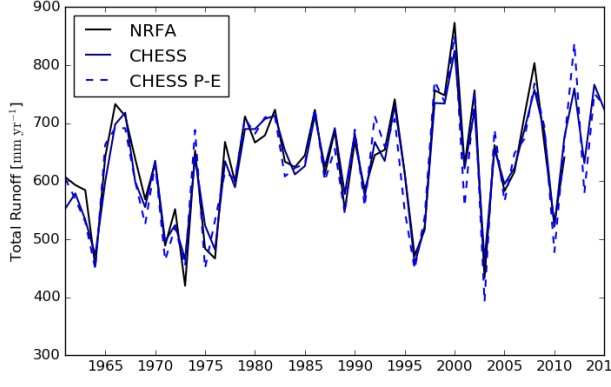



Figure 6. Comparison of CHESS annual runoff and precipitation-evaporation (P-E) to the observed GB
annual river flow (NRFA; data available up to 2011).
Fig. 7 shows the definitions of different regions used by the NRFA. They represent different GB
climate types: Scotland is wet and cold, Wales is wet and warm, England is dry and warm, English
lowlands is very dry and warm (these adjectives are relative to each other, not to a global standard).

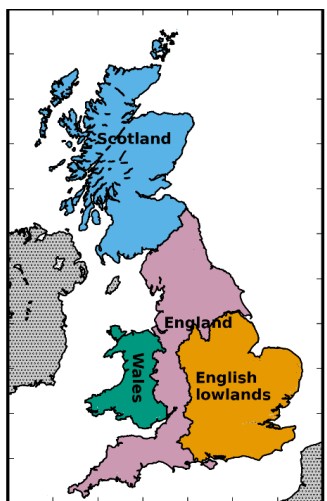

Figure 7: The four defined regions: green is Wales, blue is Scotland, orange and pink together are
England, orange alone is the English lowlands.
Fig. 8 shows the model and observation-based results for evapotranspiration in these regions, while
Table 3 gives a summary of the annual average values for the regions.





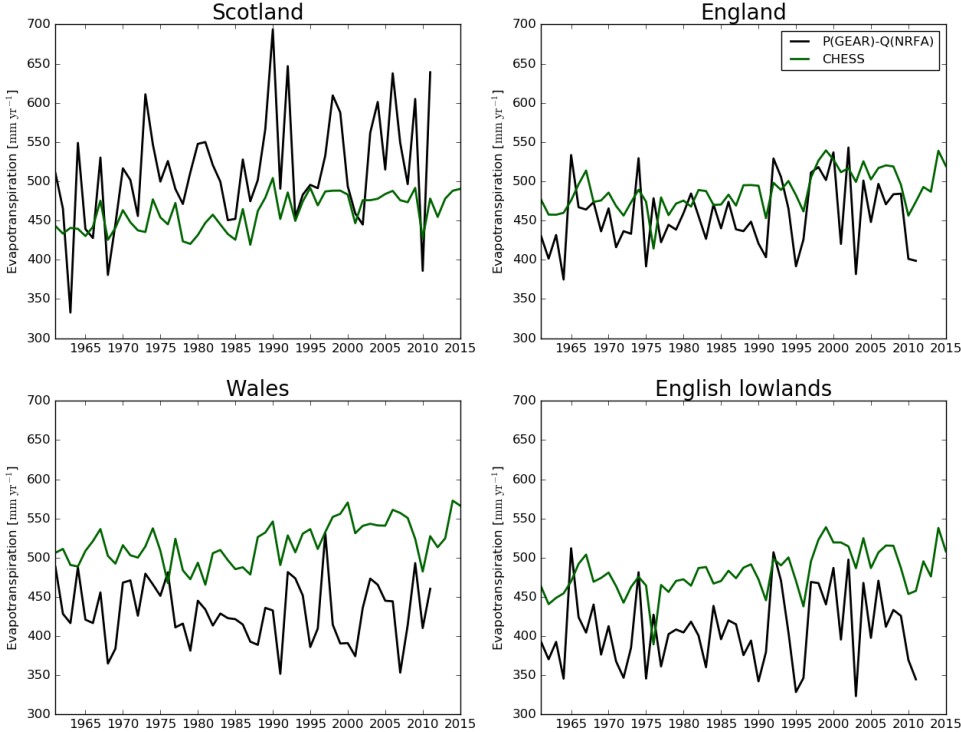

Figure 8: Annual averages for evapotranspiration in the four regions for CHESS and precipitation-runoff (P-Q) observed. Here, the observation-based products used are GEAR for precipitation and NRFA for runoff.

| Annual average (mm yr⁻¹) | Precipitation | Model Runoff | Observed Runoff | Model Evap. | Observed Evap. (P-Q) | Bias in Evap (% Observed) |
|---|---|---|---|---|---|---|
| Scotland | 1485 | 1032 | 972 | 458 | 513 | -10.5% |
| Wales | 1377 | 863 | 945 | 516 | 432 | +19.5% |
| England | 825 | 342 | 369 | 485 | 456 | +4.5% |
| English Lowlands | 681 | 206 | 273 | 479 | 408 | +17.5% |

Table 3: Annual average water balance for the four regions.

Table 3 shows that in Wales, England and the English lowlands, the model overestimates evapotranspiration by 5 to 20%. This corroborates the results of the analysis with the flux data which



show an overestimation of about 10%. However, in Scotland the comparison indicates the opposite:
the model has a lower evapotranspiration than the precipitation-runoff product. This anomalous
outcome is probably a result of the way the riverflow observations were sampled and then
extrapolated to the regional scale. As explained above, the estimate is made with 'Index
Catchments' chosen for the length of record (see Marsh et al, 2015), which in the Scottish region are
mainly in the low-lying East of the region. Due to the short record length, the Highland and West of
Scotland runoff data is not used and instead the area contribution was estimated by a model. The
evapotranspiration in this wet region will be near to the potential evapotranspiration (PET), so the
value of PET used in the model will dominate the result.
The PET product used for the gap-filling was from MORECS, which is based on observations and
calculated at the 40 km grid-scale. In flat terrain, it is a valid assumption that PET will not vary over a
40 km grid and that it can be used at the 1 km grid-scale. However, in hilly terrain such as western
Scotland, that is no longer the case. As indicated in Blyth (1999), since is it always windier on the
tops of the hills where PET is lower (due to the cold temperatures), the area-averaged PET is lower in
a hilly region than would be implied using topographic-mean data. This may explain the
overestimation of PET and underestimation of the river flows in Scotland in the observation-based
product of Hannaford (2015).
If the results for Scotland are ignored, the comparison suggests an overestimate of the model of the
order of 10 to 15%.
2.3 Evaluating the model by comparison with other modelled estimates
There are several available large-scale evapotranspiration estimates to compare to the model. All of
these estimates are model-derived. It is not expected that they will be more accurate than CHESS
and this is not an evaluation exercise. However, it is interesting to make the comparison. Table 4
describes the datasets: their derivation and the annual mean evapotranspiration for GB and the four
regions in Fig. 7. They are derived from models of differing complexity, either driven by satellite-
derived data, or ground-based weather data. Fig. 8 shows the maps of the annual mean
evapotranspiration from the four products (CHESS scaled up to 40 km, eartH2Observe, GLEAM and
MORECS) while Fig. 10 shows how the GB average varies over time.

| Product | References | Methods | Annual Evapotranspiration (mm yr$^{-1}$) | | | | |
|---------|-----------|---------|------|---|---|---|---|
| (version) | | | GB | S | W | E | EI |




| | | | | | | | |
|---|---|---|---|---|---|---|---|
| eartH2Obseve (WRR1) | Schellekens et al (2017) | Ensemble mean of 10 global models (4 land surface models and 6 hydrological models) (see Table 1 in given reference for details) | 502 | 477 | 546 | 485 | 474 |
| GLEAM (v3.0a) | Martens et al (2017) Miralles et al (2011) | Data-driven model (Priestley-Taylor), using meteorological data from reanalysis, satellite and gauged-based global datasets (0.25° resolution ) | 463 | 471 | 568 | 439 | 410 |
| MORECS (v4) | Thompson et al, (1981) Hough and Jones (1997) | Data-driven model (Penman-Monteith), using observed GB meteorological data (40 km resolution) | 524 | 482 | 543 | 528 | 522 |

1   Table 4: Description of the three large-scale evapotranspiration estimates for GB, and annual

2   averages over GB and the four regions: Scotland (S), Wales (W), England (E) and English lowlands

3   (El).

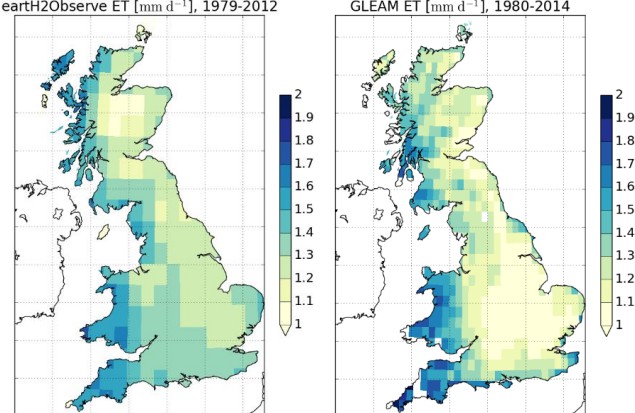



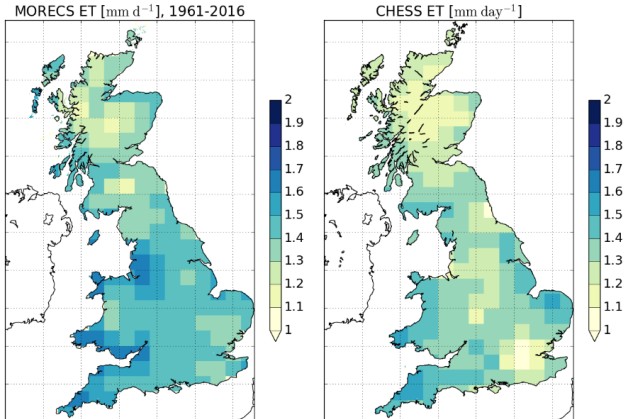

Figure 9: Maps of mean evapotranspiration for eartH2Observe, GLEAM, MORECS and CHESS (averaged up to 40 km).

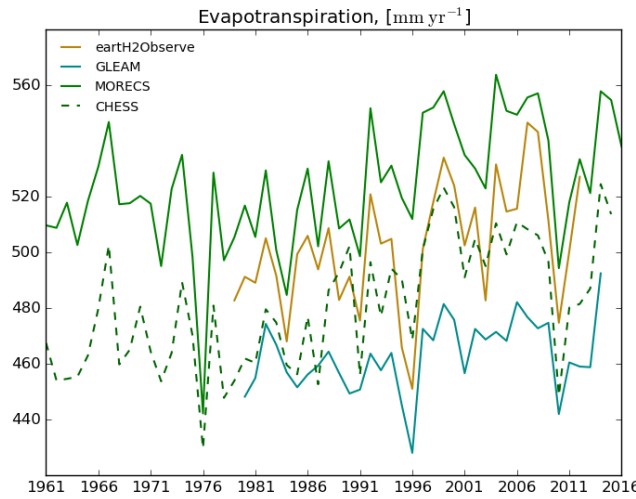

Figure 10: Annual average evapotranspiration for eartH2Observe, GLEAM and MORECS and CHESS.

It can be seen from the Figs. 9 and 10 and from the Table 4 that the main differences between the various products is the total evapotranspiration. CHESS evapotranspiration is higher than GLEAM and lower than MORECS, and close to the ensemble product from eartH2Oobserve.

To understand the evapotranspiration in all of these estimates, it is important to know the assumed role and values of PET (or equivalent) in the model. Fig. 11 shows the value of PET for GLEAM,





MORECS and CHESS (not available for E2O). It is clear from this figure that the MORECS PET is higher
than the CHESS PET, commensurate with the difference in actual evapotranspiration. This suggests
that the difference between CHESS evapotranspiration and MORECS evapotranspiration is due to
the PET. A comparison of CHESS PET with GLEAM PET shows that they have a similar magnitude.
This indicates that the reason the GLEAM evapotranspiration is lower is due to model differences,
not the PET.
This result corroborates the results of the analysis in Schellekens et al (2017) which showed that
GLEAM had much lower evapotranspiration than a wide spread of models in the temperate regions.

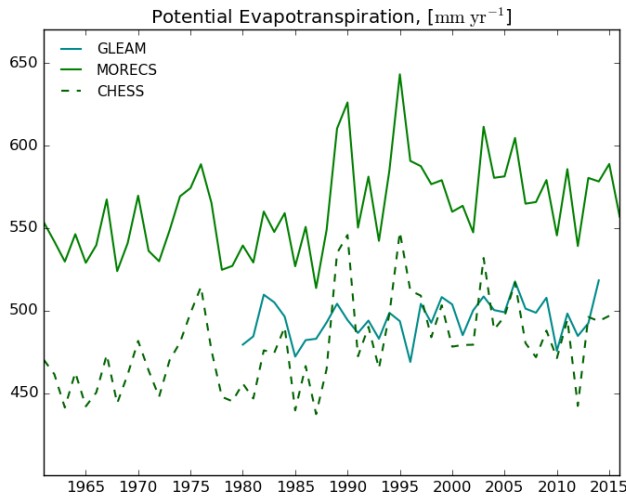

Figure 11: Annual average potential evapotranspiration for GLEAM, MORECS and CHESS.
2.4 Summary and discussions of model evaluation
The evaluation of the model show that the overall evaporation is overestimated by of the order of
10%. In winter, this overestimate is greater for trees than for grass, whereas for grass the
overestimate is in the summer. According to analysis of observations, the model underestimates the
interception from trees, especially conifer trees.
The winter evaporation for deciduous trees tends to be overestimated. This is probably due to the
known issue in the model of using a single aerodynamic resistance for the trees and underlying soil
surface evaporation.



Analysis of river flows suggests that the use of a fine grid (1km compared to 40km) for evaluating
evaporation results in more accurate estimates of the water budget over areas of high terrain.
Although comparison with observations is not perfect, the model displays reasonable allocation of
evaporation across the different components compared to observational evidence, and we
therefore deem it good enough to proceed with the subsequent analysesPETI.




3 Results
In this section, the results of the model will be used to address the four questions formulated in the
introduction.
1. Is the evaporation of GB and the regions increasing or decreasing?
2. Which components of the evaporation are contributing to the trend?
3. What meteorological changes are driving these changes?
The analysis starts with a study of annual and seasonal evapotranspiration in GB. Then we move on
to a study of evapotranspiration in the regions of GB (Scotland, Wales, England and English lowlands,
as defined in Fig. 7). These two sections answer the first question. The trends in the different
components of evapotranspiration for whole of GB are quantified to answer the second question.
The third question is addressed by studying the correlation between the evapotranspiration in the
regions and the different drivers of change (precipitation and solar radiation).
3.1 Trends in GB water balance: annual average and seasons.

| Annual (Seasons: Winter, Spring, Summer, Autumn) | Average | Rate Of Change (units yr$^{-1}$) |
|---|---|---|
| Precipitation (mm yr$^{-1}$) | 1106 (1284,896,929,1313) | 2.96±2.03 (6.95±6.25, 0.28±2.61, 2.25±4.01, 2.13±4.20) |
| Evapotranspiration (mm yr$^{-1}$) | 481 (190,585,794,345) | 0.87±0.55 (0.52±1.03, 1.12±0.64, 0.92±0.67, 0.46±0.51) |
| Runoff (mm yr$^{-1}$) | 630 (901,604,365,636) | 2.18±1.84 (3.45±5.05, 0.75±2.18, 0.86±1.67, 1.17±2.71) |
| Soil Moisture (m$^3$ m$^{-3}$) | 1267 (1345,1293,1162,1246) | -0.02±0.77 (-2.50±3.07, -0.33±0.55, 0.05±1.22, -0.02±0.98) |
| PET (mm yr$^{-1}$) | 479 (134, 592, 885, 296) | 0.74±0.66 (0.20±0.53, 1.37±0.65, 0.96±1.68, 0.38±0.49) |
| Observed Runoff (mm yr$^{-1}$) | 629 | 1.57±2.04 |

Table 5: Annual average and seasonal average fluxes and trends in the fluxes of: Precipitation,
Evapotranspiration, Runoff, Soil Moisture and Potential Evapotranspiration.



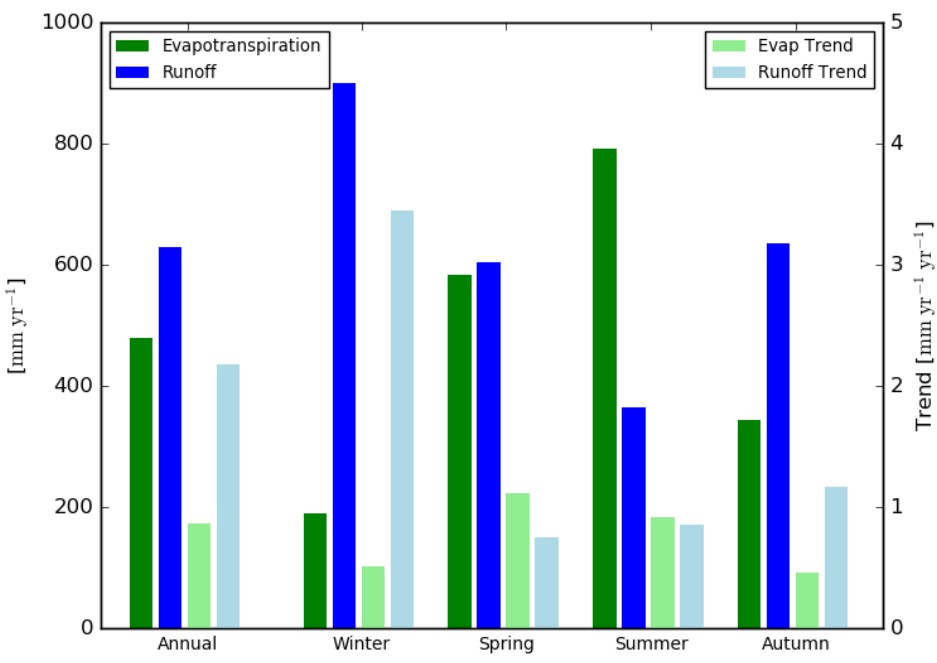

Figure 12: Bar chart showing annual and seasonal water budgets and trends (evapotranspiration and

runoff). Trends are represented on the right y-axis.

The annual average of total evapotranspiration has been presented and compared to observations in

Sect. 2 (Table 3). Here the seasonal variation in evapotranspiration is explored. The seasons are

defined as Winter (December to February), Spring (March to May), Summer (June to August) and

Autumn (September to November).

Table 5 and Fig. 12 show that here is a strong seasonality to the evapotranspiration, with summer-

time values over four times the winter-time values. This seasonality is strongly driven by the

seasonal variation in PET which has a 6-fold increase from the winter values to the summer values.

Soil moisture control reduces the summer evapotranspiration to about 75% of the PET, while the

winter and autumn evapotranspiration exceed the PET by on average 25%. The reverse is true of the

river flow which responds to a low seasonal variation in precipitation (only 1.35 variation between

winter and summer) but exhibit some soil moisture control with summer runoff 2.5 times smaller

than winter runoff.

The largest trend in evapotranspiration is seen in the spring months (1.12 mm yr$^{-1}$ yr$^{-1}$). This result

corroborates the results of an analysis of PET by Robinson et al (2017a), who demonstrated that the




largest increase in the PET for GB was in spring due to an increase of spring sunshine hours and a
decrease in relative humidity. In Sect. 4 we discuss the impact of this on the riverflow.
Unlike the PET results however, the smallest trend of evapotranspiration is seen in the autumn. This
may be due to soil moisture control of evapotranspiration.
The trends in runoff are overall two times larger than the evapotranspiration trends. This is due to
the very large increase in winter runoff due to the large positive trend in precipitation in Scotland.
The other seasons have a similar upward trend of about 1 mm $yr^{-1}$ $yr^{-1}$.
3.2 Regional trends.

| Annual Average (rate of change in units $yr^{-1}$) | Scotland | Wales | England | English lowlands |
|---|---|---|---|---|
| Precipitation (mm $yr^{-1}$) | 1495 (5.41±3.00) | 1386 (2.93±2.86) | 832 (1.51±1.87) | 686 (1.07±1.76) |
| Evapotranspiration (mm $yr^{-1}$) | 460 (0.87±0.47) | 518 (0.93±0.72) | 487 (0.86±0.66) | 481 (0.91±0.71) |
| Runoff (mm $yr^{-1}$) | 1042 (4.56±2.82) | 869 (2.12±2.80) | 347 (0.79±1.75) | 209 (0.33±1.50) |
| PET (mm $yr^{-1}$) | 423 (0.53±0.46) | 496 (0.69±0.68) | 509 (0.87±0.77) | 519 (1.03±0.86) |
| Soil Moisture ($m^3$ $m^{-3}$) | 1534 (0.29±0.56) | 1139 (-0.03±0.66) | 1128 (-0.20±1.00) | 1177 (-0.29±1.12) |
| Observed total runoff (mm $yr^{-1}$) | 972 (3.60±2.63) | 945 (2.67±3.72) | 369 (0.38±1.83) | 273 (0.13±1.68) |

Table 6: Annual average GB and regional fluxes and trends in the fluxes of: Precipitation,
Evapotranspiration, Runoff, Potential Evapotranspiration and Soil Moisture. Bottom row shows
annual observed averages from NRFA runoff product.

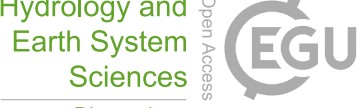

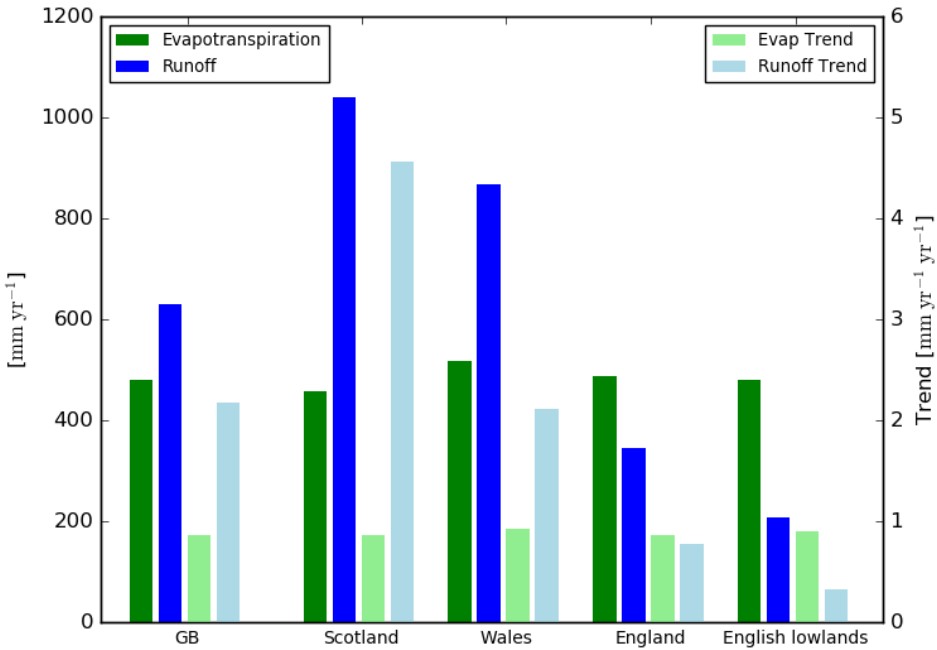

Figure 13: Bar chart showing annual GB and regional water budgets and trends (evapotranspiration
and runoff). Trends are represented on the right y-axis.
The water balance of the different regions can be summarised as follows: the water balance is runoff
dominated in Scotland (runoff is 70% of rainfall) and evaporation dominated in the English lowlands
(runoff is 30% of rainfall).
Table 6 and Fig. 13 show that the trend in evapotranspiration is roughly constant between regions,
in the same way that the total evapotranspiration is fairly constant. However, the runoff trends are
more varied between regions. Scotland dominates the trend with an increase in runoff of 4.56 mm
$yr^{-1} yr^{-1}$ and the English lowlands have the smallest trend of 0.33 mm $yr^{-1} yr^{-1}$.
3.3 Totals and trends in the components of evapotranspiration for GB.
There are three components of the evapotranspiration. The JULES model calculates each explicitly
and their totals and trends for GB are given in Table 7.

| | Average (mm $yr^{-1}$) | Rate Of Change (mm $yr^{-1} yr^{-1}$) |
|---|---|---|
| Evapotranspiration | 481 | 0.87±0.55 |





| | | |
|---|---|---|
| Soil surface evaporation | 100 (21%) | 0.14±0.47 (16%) |
| Transpiration | 229 (48%) | 0.47±0.23 (51%) |
| Interception | 137 (28%) | 0.31±0.22 (33%) |

Table 7: Annual average and trends in GB evapotranspiration and components.
The percentage of evapotranspiration that is due to interception is about 30%, due to soil surface
evaporation is about 20% and due to transpiration is due to about 50%. Despite the possible errors
in the modelled allocation of evapotranspiration between the components reported in Sect. 2, it is
likely that the trends of the components are well represented. A key factor is the contribution of the
component to the trend in total evapotranspiration. These vary between components and they are
slightly different to the contributions they make to the annual average. This means they are not all
increasing at the same rate: the interception in increasing more rapidly than the other two
components. This is discussed in more detail in Sect. 4.
Figure 14 displays how the components vary across the regions. Although transpiration generally
dominates, they are nearly equal in Scotland which experiences very high rainfall rates.

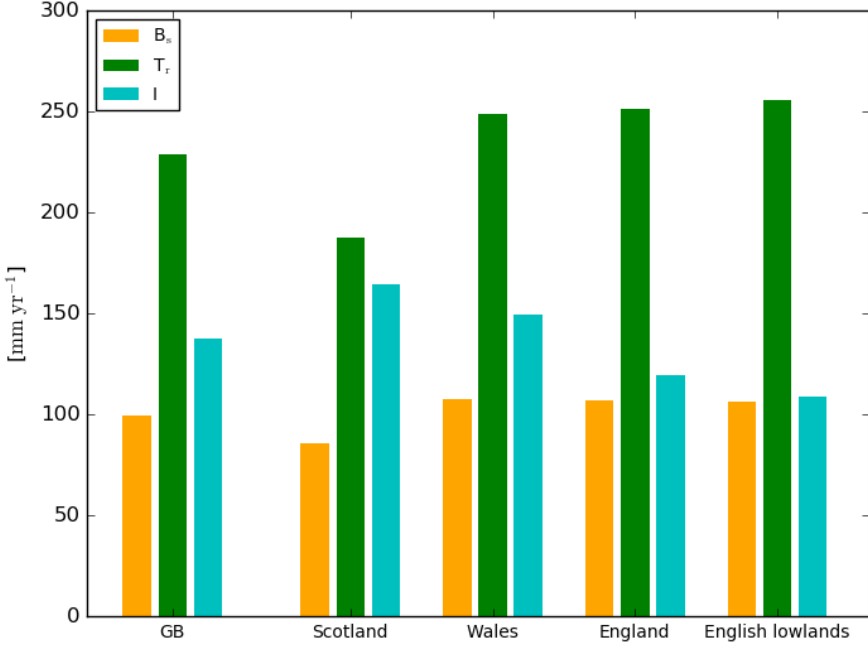

Figure 14: Bar chart of annual average evapotranspiration components in GB and regions.




### 3.4 What meteorological variable is driving the trend?
To understand the reason for the trend in evapotranspiration, we study the correlation between the
annual evapotranspiration and its two main drivers: precipitation and radiation. The analysis is
carried out for the regions and for the separate components. A discussion of the trends in radiation
due to changes in aerosols and cloud cover and how it is represented in the CHESS dataset is given in
Sect. 4.
Table 8 and Fig. 14 show the results of the correlations.

| R | $P$ v $E_{tot}$ | $S_w$ v $E_{tot}$ | $P$ v $I$ | $S_w$ v $I$ | $P$ v $T_r$ | $S_w$ v $T_r$ | $P$ v $B_s$ | $S_w$ v $B_s$ |
|---|---|---|---|---|---|---|---|---|
| GB | 0.66 | 0.42 | 0.86 | -0.09 | 0.11 | 0.74 | 0.49 | 0.44 |
| Scotland | 0.65 | 0.50 | 0.80 | -0.01 | 0.09 | 0.81 | 0.51 | 0.44 |
| Wales | 0.47 | 0.45 | 0.83 | -0.15 | -0.20 | 0.84 | 0.27 | 0.37 |
| England | 0.58 | 0.28 | 0.84 | -0.24 | 0.12 | 0.56 | 0.31 | 0.42 |
| English lowlands | 0.64 | 0.19 | 0.82 | -0.28 | 0.35 | 0.36 | 0.25 | 0.46 |

Table 8: Correlation of annual Precipitation (P) and Shortwave radiation ($S_w$) with Total
evapotranspiration ($E_{tot}$), Interception (I), Transpiration ($T_r$) and Soil surface evaporation ($B_s$), for GB
and for the four regions.

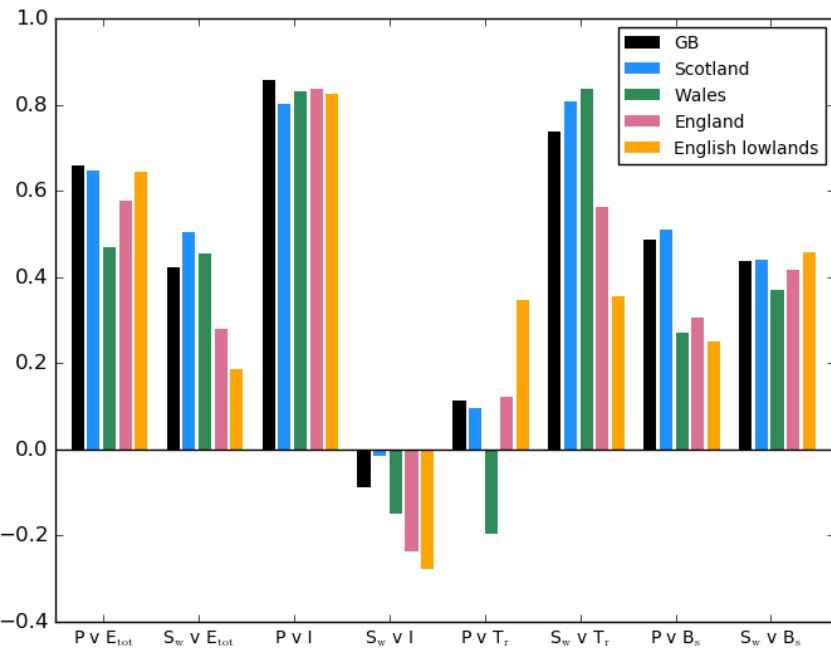





Figure 15: Bar chart showing correlation of annual Precipitation (P) and Shortwave radiation ($S_w$)
with Total evapotranspiration ($E_{tot}$), Interception (I), Transpiration ($T_r$) and Soil surface evaporation
($B_s$) for GB and the regions.
The important take-away result from this Fig. 15 and Table 8 is that there is always a strong
correlation between rainfall and interception (3rd column in Fig 15 and Table 8) and always a strong
correlation of transpiration with radiation (6th column in Fig. 15 and Table 8).
From the analysis, soil surface evaporation seems to have similar correlations with both precipitation
and radiation.
The implications for these results is discussed in the following section.

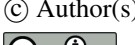



4. Discussion
4.1 Trends in the annual GB water budget
Hannaford (2015) suggest that the overall runoff of GB is increasing by 1.6 mm yr$^{-1}$ yr$^{-1}$ while
Robinson et al (2017a) show an increase in precipitation of 2.86 mm yr$^{-1}$ yr$^{-1}$ (updated analysis of
data including later years shows 2.96 mm yr$^{-1}$ yr$^{-1}$ – Table 5) If this was taken as the only evidence for
a change in actual evapotranspiration, we could calculate an increase of 1.36 mm yr$^{-1}$ yr$^{-1}$. However,
Robinson et al (2017a) indicate an increase in PET, which represents an upper limit of transpiration,
of only 0.77 mm yr$^{-1}$ yr$^{-1}$ and Kay et al (2013) have suggested that the overall trend in
evapotranspiration of GB is close to the trend in PET, which they estimate as 0.7 mm yr$^{-1}$ yr$^{-1}$.
If all this were true, then there would be an increase in soil moisture or groundwater store of the
order of 0.5 mm yr$^{-1}$ yr$^{-1}$.
The results of this study give slightly different results. The overall trend in evapotranspiration is 0.87
mm yr$^{-1}$ yr$^{-1}$ and the runoff is larger than that given by Hannaford (2015) at 2.16 mm yr$^{-1}$ yr$^{-1}$. This
then almost balances the water budget leaving negligible increase in soil moisture or groundwater.
We have already demonstrated (Sect 2.2.4) that the Hannaford (2015) estimates of riverflow are too
low in Scotland. This was probably due to the use of large-scale area-average Potential Evaporation
over hilly terrain to gap-fill the data which will tend to overestimate the evaporative loss leading to
runoff-estimates that are too low. It is possible that the trend in runoff, rather than the total, is
affected by the lack of interception trend (discussed below) in the estimated runoff. A deeper
analysis of the flows in Scotland should be carried out to understand these trends.
The reason why the evaporative loss might be greater than the potential could be due to the large
fractional component of interception (30%): in the wet and windy areas (West Scotland), there is a
higher fraction of evergreen needle leaf trees which have a high interception capacity (see Calder,
1990). The evaporation from a wet forest often exceeds the PET, drawing down energy in the form
of negative sensible heat (i.e. cooling the air) to drive it (Stewart et al, 1977).
If the drivers of interception are increasing faster than the PET, then with such a high interception
fraction, there might be a larger trend in the evapotranspiration than the potential evaporation. This
is discussed in Sect. 4.4.
4.2 Impact of seasonal variation of trend in evapotranspiration on river flows.



The results of this study indicate a clear increasing trend in spring evapotranspiration, driven by an
increase in PET reported in Robinson et al (2017a).
It is interesting to note the impact of this on the river flows. In their review, Watts et al (2015)
suggest that there is no clear evidence of how actual evaporation has changed over the last few
decades. Hannaford (2015) notes that, while there is an observed increase in winter, summer and
autumn river flows from 1961 to 2010, there is a decrease in the spring (see table 1). This is
reiterated in Harrigan et al., (2017) who note that we need an 'improved understanding of the
drivers of these seasonal changes in river flow'.
This study clearly indicates that some of the spring-time drop in runoff will be due increased
evapotranspiration in addition to a reduction of rainfall.
4.3 Temporal changes in shortwave radiation over the study time period in Great Britain.
There is an overall increase in downward short wave radiation apparent in the observations used in
this study of 1.1±0.7 Wm$^{-2}$ decade$^{-1}$ over GB.
There are two possible reasons for an increase in short wave radiation: one is that there is less cloud
and the other that the air is more transparent due to a reduction in aerosols (pollution).
The aerosol effect has been widely studied and is referred to as 'dimming and brightening' since, the
aerosol amount increased throughout the 20th century until about the 1970's, when due to
legislation, it began to fall. Observations reported by Wild (2009) illustrate this at the global scale.
Across Europe, the dimming resulted in a decrease in the total Short Wave Radiation of 1.4 Wm$^{-2}$
decade$^{-1}$ from the beginning of the century to the mid-1970s and the brightening to an increase of
2.2 Wm$^{-2}$ decade$^{-1}$ up to the mid-1990s. This change in radiation has been shown to have a
significant impact on river flows in the region (Gedney et al 2014).
GB has a somewhat different temporal pattern of dimming and brightening to mainland Europe.
Using an ensemble of computer model runs, Folini and Wild (2011) simulated the aerosols in Europe.
Their results agree with the overall result of Wild (2009) for Europe, but also note some interesting
regional variations: in the case of GB from 1960 onward, there is no dimming, only brightening. For
the All-Sky conditions (i.e. including clouds. See Fig 9 of the Folini and Wild (2011) paper) the
increase in radiation is about 0.8 Wm$^{-2}$ decade$^{-1}$. This tallies with an observational study by Stjern et
al. (2009) which included a long term record at Aberdeen (the only GB station in the study) where
there was no dimming from 1960's only a brightening of 1 Wm$^{-2}$ decade$^{-1}$. This is also consistent with
the increase of short wave (0.9 ±1.1 Wm$^{-2}$ decade$^{-1}$ for the years 1979-2012) given in the





observation-based meteorological forcing dataset WFDEI (Weedon et al, 2014) which included
explicit aerosol effects.
The short-wave radiation data used in CHESS is based on sun-shine hours. There is a spatially varying
aerosol effect included, but it is based on data from 1978 and does not change in time. As reported
in Robinson et al (2017), it may be that the aerosol effect is implicit in the sunshine hours record as
there is a threshold of 120 Wm$^{-2}$ and the number of hours above that threshold will be affected by
the pollution levels, although the changes in cloud cover are likely to be dominant in this signal.
The biggest increase in short wave is in the spring. This is consistent with data from across Europe
(Sanchez-Lorenzo et al., 2008) and may be due to changes in the Atlantic Multidecadal Oscillation
(AMO) which has resulted in decreased spring precipitation in Northern Europe (Sutton and Dong,

11   2012).

It is concluded therefore that the CHESS data gives a reasonable representation of the shortwave
radiation over GB for the study period.
4.4 Analysis of correlations.
It is generally accepted (e.g. Teuling et al., 2009, Wang and Dickenson, 2012), that in moisture
limited regions, there will be a strong correlation of evapotranspiration with precipitation, while in
wet, cloudy regions, there will be a stronger correlation with radiation. However, in the analysis
presented by Teuling et al (2009), most of GB is deemed to be radiation limited (confirming its wet
and cloudy climate) apart from west Scotland and central England. The result for west Scotland is a
counter-intuitive result since it is the wettest region of GB.
A correlation analysis of annual evapotranspiration with precipitation and short-wave radiation was
carried out with the CHESS model results and reported in Sect. 3.4. In this analysis, the same
patterns emerged: the correlation of evapotranspiration and precipitation was high in Scotland. To
understand why the evapotranspiration correlates more strongly with precipitation than radiation in
Scotland, the correlations for the four regions for each of the components of evapotranspiration
against precipitation (P) and shortwave ($S_w$) radiation were made.
The important take-away result from this analysis (see Fig. 14 and Table 8) is that there is always a
strong correlation between rainfall and interception (3$^{rd}$ column in Fig 14 and Table 8). This tallies
with the results of the analysis in Sect. 4 and the evidence from Nisbet (2005) that interception is
near to a fixed fraction of rainfall. Interception is not 'limited' by rainfall, but it strongly correlates
with it because it is unlimited.





Transpiration however is generally limited by energy in GB since there is usually enough water and
often not enough sunshine. This results in a strong correlation of transpiration with radiation (6[th]
column in Fig. 14 and Table 8).
Soil surface evaporation has similar correlations with both precipitation and radiation.
These results show why the evapotranspiration-precipitation correlation dominates in Scotland: it is
because the rainfall is high and so interception is nearly as large as the Transpiration component of
the evapotranspiration (see Fig. 15). English lowlands provide the opposite case. The
evapotranspiration is dominated by transpiration (see Fig. 15) and it is equally water limited and
radiation limited.
4.5 Importance of interception to the GB hydrology.
As discussed above, this study highlights the importance of interception to the overall hydrological
balance and trends in GB. It may also have implications for floods.
Floods represent a major hazard for the UK and recently there has been effort to identify natural
ways to alleviate the floods. Dadson et al (2018) and Stratford et al (2017) both review the evidence
of whether the presence of trees have an impact to reduce flood peaks. They both conclude that
there can be an impact of mature trees to reduce smaller floods, probably do the interception.
Using a physically based model (JULES) with detailed representations of all the evaporation
components, including interception, enables us to quantify the effect of land-cover on hydrology at
the GB scale. The interception in the model has some uncertainties, and possibly underestimates the
effect (see Sect. 2.2.3). However, the results are consistent with the observations and can be used as
evidence for this increasingly important issue.
4.6 Possible impact of rising $CO_2$ levels on water balance of GB.
Over the 55 years of the study period, there has not only been a change in the physical climate, but a
50% increase in the atmospheric concentration of $CO_2$, from 300 ppm to 450 ppm. It is possible that
this has affected the water balance of GB due to the response of vegetation to $CO_2$. There is
evidence (Leakey et al, 2009) at the canopy scale that plants will transpire less in increased
atmospheric $CO_2$ levels. However, models differ substantially in their prediction of the large-scale,
long term response to elevated $CO_2$ which includes changes to ecosystems and vegetation structure
(de Kauwe et al, 2013).





The JULES model includes the effect of an increase in $CO_2$ on evapotranspiration, although it might
reduce the transpiration too much (Prudhomme et al, 2014). As a result of this uncertainty, in the
version of CHESS reported here, we used the $CO_2$ level half way through the period (in 1986) and
kept it constant.
However, it is instructive to analyse to what extent the water balance would be predicted to change
in the presence of a 50% increase in $CO_2$ according to the model. The model showed that overall
evapotranspiration and runoff were unchanged. There was a change in the trend however:
evapotranspiration trend was reduced: to 0.60 mm $yr^{-1}$ $yr^{-1}$ (from 0.87 mm $yr^{-1}$ $yr^{-1}$) while the positive
trend in runoff increases: to 2.33 mm $yr^{-1}$ $yr^{-1}$ (from 2.16 mm $yr^{-1}$ $yr^{-1}$ with no $CO_2$ increase).  Most of
the difference in trend comes from a lower positive trend in transpiration of 0.11 mm $yr^{-1}$ $yr^{-1}$
(compared to 0.47 mm $yr^{-1}$ $yr^{-1}$ in the constant $CO_2$ run). In the earlier part of the run, when transient
$CO_2$ levels were lower than the constant $CO_2$ run, the stomata were more open, so stomatal
conductance was higher, while the reverse was true in the later part of the run, with higher $CO_2$
levels. This leads to the same mean transpiration over the run, but a weaker trend.



5. Conclusions
This study set out to explore the long term evolution of the water budget of Great Britain (GB),
including how and why it is changing.
A study of the river flows (summarised in Hannaford, 2015) suggest that the runoff from GB is
increasing with time (1.6 mm yr$^{-1}$ yr$^{-1}$ over the last 50 years) with a decrease in spring time (Harrigan
et al, 2017). It is possible that the increase is an underestimate due to the methods of gap-filing
used.
The Potential Evapotranspiration (PET), which expresses the likely increase of a well-watered grass
due to changes in wind, radiation and temperature, has been estimated by Robinson et al (2017a) to
be 0.77 mm yr$^{-1}$ yr$^{-1}$ and by Kay et al (2013) to be 0.7 mm yr$^{-1}$ yr$^{-1}$.
In order to examine the actual evapotranspiraiotn, we used a validated model of the land surface
(JULES: Joint UK Land Environment Simulator) which was driven by observation-based
meteorological data (Robinson et al, 2017a) over 5 decades. The modelled evapotranspiration
increased at a rate of 0.87 mm yr$^{-1}$ yr$^{-1}$ which is greater than the increase in Potential
Evapotranspiration.
The analysis showed that the interception was a large component of the overall evapotranspiration
(30%) and increasing at a slightly higher rate (proportionally) than the other components.
In this paper, it was demonstrated that annual interception correlates strongly with annual
precipitation rather than solar radiation. Over the last 5 decades, precipitation has increased faster
(2.96±2.03 mm yr$^{-1}$ yr$^{-1}$) than the PET (0.74±0.66 mm yr$^{-1}$ yr$^{-1}$).  This increase in precipitation,
combined with the high interception rates in GB (due to the wet conditions) explains why the trend
in evapotranspiration is higher than the trend in PET.
Thus the paper clearly demonstrated that a representation all three components of
evapotranspiration (interception, transpiration and bare-soil evaporation) are needed to understand
the trends in the GB water-budgets.
In addition, the study demonstrated the importance of evapotranspiration to the annual and
seasonal water budget of Great Britain and its regions. For instance, the observed decreasing spring-
time runoff reported in Harrigan et al (2017) is a combination of decreased in the precipitation and
increased evaporation in this season.





Future directions for this research should include a focus on the evaporation processes that have
been demonstrated to be so important: interception, the winter-time soil surface evaporation under
deciduous trees and the summer-time transpiration of grass. Narrowing our uncertainty in these key
processes will help identify the role of land cover on the UK water budgets.



6. Data and model availability
This paper presents the results of the model in a particular code version and configuration. The
version of a model refers to the code which are used to solve the equations of the processes (this
can be summarised by its Version Number). The configuration includes the selection of the options
used (such as whether to use dynamic vegetation or not), the look-up tables of parameters used in
the equations, the meteorological data used to drive the model and the ancillary data used to set up
the maps of soils and land cover classification. This will also have a Version Number and a reference
name. This information is given in Appendix B.
Code availability: The JULES code is available freely and can be applied for through the JULES
repository: https://code.metoffice.gov.uk/trac/jules. The version used for the CHESS runs based
on JULES vn4.5, branch: r3488_albmar_spdm. More detailed information about the JULES code,
use and availability: http://jules.jchmr.org.
Data availability: The data (drivers and initial inputs) of the Configuration for running the model for
CHESS is described in the rose suite number u-au394 (see Appendix B). All the model outputs
analysed here and more (evaporation components, runoff, latent and sensible fluxes, soil
moisture and temperature, surface temperature, carbon gross and net productivities) are
publicly available as the CEH CHESS-land dataset (Martinez-de la Torre et al, 2018b).
The flux data used for evaluation is available upon request from the author.





Appendix A
A1 Overview of hydrology in JULES
This summary follows water as it arrives as precipitation at the land surface, and describes the
journey it takes through the land surface through interception, the snow pack, surface runoff
generation, vertical adjustment of soil moisture, drainage and evapotranspiration (Best et al, Clark et
al).
Each of the above-ground processes (interception, snow and evapotranspiration) are calculated for
each of the different land-surfaces represented within each grid-box. While the equations are
universal, the parameters change for each land cover type. The derivation of the maps of fractional
cover of each land cover and the parameters used are described in Appendix B.
The below ground processes (surface runoff generation, vertical adjustment of soil moisture and
drainage) are calculated based on the grid-average soil moisture. Parameters for these equations
depend on the soil type and topographical information. The derivation of these for CHESS are given
in Appendix B.
Since this application is in the UK which has a temperate climate, we have not covered the cold
processes included in JULES, such as snow and soil freezing. Snow sublimation only accounts for 1%
of the modelled total evapotranspiration for the GB area (2% in Scotland, 0.5% in the rest of GB) and
so, for the sake of brevity, we have omitted to describe the snow model here. Readers are referred
to Best et al (2011) for more details.
A2. Rainfall intensity and interception
One of the most important aspects of any distributed hydrology model is how to deal with
precipitation that is supplied as a time- and space- average. If the precipitation is assumed to fall
evenly over a wide area or over a long time period, then infiltration through the vegetation canopy
or top soil layers is too low, resulting in a surface that is too wet and deeper soils that are too dry.
In JULES, the spatial distribution of intensity of rainfall is assumed to follow an exponential statistical
distribution.
$$f(P) = \left(\frac{\mu}{P}\right) \exp\left(\frac{-\mu P_i}{P}\right)$$    (A1)




where $P$ (kg m$^{-2}$ s$^{-1}$) is the area-average rainfall rate, $P_i$ (kg m$^{-2}$ s$^{-1}$) is the rainfall rate over a small area
and $\mu$ is the fraction of the grid box area over which the rain is assumed to fall. In CHESS, since the
grid box is small, this is set as 1.
This rainfall-intensity distribution is used in two ways. Firstly to calculate how much of the rainfall
falls through the vegetation canopy to the understorey, called 'throughfall' ($T_f$). Secondly, how much
of the throughfall infiltrates the soil (see Sect. A3). The throughfall is dependent not just on the
rainfall intensity but how much water is already on the leaves and also has a dripping element to it.

$$T_f = P\left(1 - \frac{C}{C_m}\right)exp\left(-\frac{\varepsilon C_m}{P\Delta t}\right) + P\frac{C}{C_m} \qquad (A2)$$

where $C$ (mm) is the amount of rainfall stored on the leaves, $C_m$ (mm) is the maximum capacity
which depends on the leaf area index of the vegetation and $\varepsilon$ is a tuning factor.
Vegetation is assumed to have a certain capacity that gets filled up by rain and reduced by
evaporation, called 'interception'. The interception rate is calculated as a fraction of the open-water
evaporation rate, where the fraction ($F$) is the proportion of water stored to the maximum capacity
of that vegetation type.

$$F = \frac{C}{C_m} \qquad (A3)$$

The value of $F$ (dimensionless) is converted to an effective surface resistance which is merged with
resistances from the other evaporation components for the calculation of the tile-
evapotranspiration based on the surface energy balance (see Sect. A6).
A3. Surface water partitioning
Surface runoff is generated either through infiltration excess (also known as Hortonian flow), or
saturation excess (also known as Dunn flow). These are both represented in the JULES model.
Infiltration excess runoff will be generated by JULES if the water-flux over the time step (either
rainfall, throughfall or snow melt) exceeds the maximum infiltration rate of the soil. The water-flux
rate is assumed to vary in intensity across the grid and to have been altered by the interception (see
Eq. (48) in Best et al, 2011). The maximum infiltration rate is the saturated hydraulic conductivity
multiplied by a vegetation dependent parameter (4 for trees and 2 for grasses). However, this
maximum infiltration rate is so high that it is never invoked.
Saturation excess runoff is based on the concept of the fact that the soil moisture across an area will
not be uniform but that a fraction of the model grid cell ($f_{sat}$) will be at saturation. This fraction is





used as a multiplier on the remaining water arriving at the soil surface to convert to surface runoff.
There are two options for representing this in JULES. In this study we use the PDM, based on a
Pareto distribution to describe the spatial variation of soil moisture:

$$f_{sat} = 1 - \left(1 - \frac{\theta - \theta_0}{\theta_s - \theta_0}\right)^{b/b-1} \qquad \text{(A4)}$$

where $\theta$ (m$^3$ m$^{-3}$) is the mean volumetric soil moisture, $\theta_0$ (m$^3$ m$^{-3}$) is the minimum soil moisture for
the PDM scheme to start producing surface runoff, and $\theta_s$ (m$^3$ m$^{-3}$) the saturated soil moisture. *b*
(dimensionless) is a tunable parameter. Martinez-de la Torre et al (2018a) introduced a method for
identifying values of *b* and $\theta_0$ from topographical data in the UK and these values are used here.
A4. Vertical soil water distribution
After the interception and the surface runoff, the remaining water enters the soil at the surface and
is redistributed through the 3 m of soil moisture using the Darcy-Richard Equations. The resulting
soil moisture profile allows the model to distinguish between the soil moisture near the surface
which controls the soil surface evaporation and deeper layers which can be accessed by plants with
different root depths (see Sect. A6). This distinction affects the sub-diurnal timing of energy fluxes
which is important for weather prediction. The essential equations are as follows:

$$W = k\left(\frac{d\psi}{dz} + 1\right) \qquad \text{(A5)}$$

where *W* (kg m$^{-2}$ s$^{-1}$) is the vertical flux of water through the soil, z (m) is the vertical distance, *k* (kg
m$^{-2}$ s$^{-1}$) is the conductivity of the soil, and $\psi$ is the suction (m). There are two options for calculating
the conductivity and suction in JULES. In this study we use the van Genuchten (1980) formulations:

$$\left(\frac{\theta}{\theta_s}\right) = \frac{1}{\left[1 + (\alpha\psi)^{\left(\frac{1}{1-m}\right)}\right]^m} \qquad \text{(A6)}$$

$$k = k_s \left(\frac{\theta}{\theta_s}\right)^{0.5} \left[1 - \left(1 - \left(\theta/\theta_s\right)^{1/m}\right)^m\right]^2 \qquad \text{(A7)}$$

Where $\psi_s$ (m) is the suction at saturation and $k_s$ (kg m$^{-2}$ s$^{-1}$) is the conductivity at saturation while $\alpha$
and *m* are model parameters. All four of these parameters are dependent on the soil type and the
values used in CHESS are described in Appendix B.
Not shown here are the heat-diffusion equations which act on the same layers. The transfer of heat
is important to the soil hydrology when the soil freezes which alters the hydraulic conductivity.




A5. Drainage
The bottom boundary condition of the soil column can affect the performance of the model. At 3 m
deep in the standard version, the model is assumed to drain at the rate of the gravity drainage using
the soil moisture of the bottom layer – thus assuming there is an infinitely thick layer below with the
same soil moisture. Different options are available such as a zero flux layer or a deeper groundwater
store that can bring water upwelling and are presented in Best et al (2011).
A6. Evapotranspiration
Apart from the interception (see Sect. A1), the evaporation is assumed to consist of two
components: bare soil evaporation and transpiration.
The soil surface evaporation control is the simplest and depends on the soil moisture in the top
layer:

$$r_{soil} = 100 \left(\frac{\theta_c}{\theta_1}\right)^2 \tag{A8}$$

Where $r_{soil}$ (s m$^{-1}$)is the surface resistance used for bare soil, $\theta_1$ (m$^3$ m$^{-3}$) is the soil moisture in the top
soil layer and $\theta_c$ (m$^3$ m$^{-3}$) is the soil moisture at critical point (defined in Appendix B).
The transpiration is more complex as it depends on the response of the photosynthesis of the plant
to environmental controls: light levels, temperature, ambient carbon dioxide, soil moisture and
humidity. The surface resistance of the plant to the optimum photosynthesis (no limits) is inferred
and assumed to apply to the transpiration.
The effective surface resistance is made from a combination of the three components: interception,
bare soil and transpiration. The actual evapotranspiration depends on the amount of energy
available, which depends on the radiation balance and the soil heat fluxes. Details are given in Best
et al (2011).



Appendix B
The configuration (ancillary files, science options and parameters, and driving data) used for the
CHESS simulation presented here are all documented in the rose suite u-au394
(https://code.metoffice.gov.uk/trac/roses-u/browser/a/u/3/9/4/). Instructions for how to access
this are given in the website: http://jules.jchmr.org. A summary of the options and driver datasets is
given here.
The model is run with 8 surface tiles (5 plant functional types: broadleaf trees, needleleaf trees,
grasses, shrubs and crops; and 3 non-vegetated types: open water, bare soil and urban) derived
from the CEH Land Cover 2000 (Fuller et al, 2002) map at a resolution of 25 m. Table B1 shows how
the Land Cover 2000 classes were mapped onto JULES land cover types. These were then aggregated
to 1 km resolution, as fractions of the total gridbox.
The soil hydrology component of JULES is based on the Darcy Richards equations (see Appendix A for
a summary) and, in this configuration, the van Genuchten (1980) approach, with the vertical
discretization into four layers of varying depth: 0.0-0.1 m, 0.1-0.35 m, 0.35-1.0 m and 1.0-3.0 m.  The
soil hydraulic characteristics are assumed to be spatially uniform for each grid cell, and have been
calculated for the model domain from the Harmonised World Soil Database (HWSD;
FAO/IIASA/ISRIC/ISS-CAS/JRC, 2012), by classifying soils by their texture, then using the values from
Wösten et al (1999). We used a newly developed terrain slope dependency for the PDM scheme in
the production of saturation excess runoff (Martinez-de la Torre et al, 2018a), and the slope was
derived from the Great Britain 50 m resolution CEH-IHDTM database (Morris and Flavin, 1990;

21    1994).

The phenology for each month was prescribed for the deciduous vegetation and the crops. The
dynamical vegetation scheme was switched off. A 10-layer approach is used for canopy radiation
interception, including an exponential decline of leaf nitrogen with canopy height. For the run with
increasing atmospheric $CO_2$ concentration, we used annual values from the US NOAA (National
Oceanic and Atmospheric Administration) Global Monitoring Division
(https://www.esrl.noaa.gov/gmd/ccgg/trends/global.html).
A 10-year spin up run was conducted to initialize the model. The meteorological data used to drive
the model is publicly available (CHESS-met: Robinson et al, 2017b). It is a daily dataset based on
observations from 1961 to 2015. The model is integrated at a half-hourly time step, using a daily
disaggregation scheme (Williams and Clark, 2014) to disaggregate the driving data. In terms of
precipitation, precipitation events start at a random time during the day and last for 2 hours in the





case of convective precipitation, or 5 hours in the case of large-scale precipitation. The input rainfall
is assumed to be convective for temperatures above 15 °C, and covers the complete 1 km grid
cell.The rest of options and parameters (radiation, snow, vegetation, non-vegetated tiles) are
available on the rose suite u-au394.

| Land Cover 2000 class | JULES surface type |
| --- | --- |
| Water (inland) | Water |
| Saltmarsh | Water |
| Supra-littoral rock | Bare soil |
| Supra-littoral sediment | Bare soil |
| Bog (deep peat) | Shrub |
| Dense dwarf shrub heath | Shrub |
| Open dwarf shrub heath | Shrub |
| Montane habitats | Shrub |
| Broad-leaved / mixed woodland | Broadleaf tree |
| Coniferous woodland | Needleleaf tree |
| Improved grassland | Grass |
| Neutral grass | Grass |
| Setaside grass | Grass |
| Bracken | Shrub |
| Calcareous grass | Grass |
| Acid grassland | Grass |
| Fen, marsh, swamp | Grass |
| Arable cereals | Crop |



| Arable horticulture | Crop |
|---|---|
| Arable non-rotational | Shrub |
| Suburban / rural development | Urban |
| Continuous urban | Urban |
| Inland bare ground | Bare soil |

Table B1 Allocation of Land Cover 2000 map classes to JULES surface types
Author Contribution: EMB designed the study, carried out the analysis and wrote the manuscript,
AM performed the model runs, summarised the results and created the figures, ELR set up the
system for running the model (ancillary data and driving data) and for evaluating the model with the
flux data and runoff data.

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
