# Peer review of "Trends in evapotranspiration and its drivers in Great Britain: 1961 to 2015"

_Hydrology and Earth System Sciences, 2018_

## Referee Comment (RC1) · Anonymous Referee #1 · 12 May 2018

**Trends in evapotranspiration and its drivers in Great Britain: 1961 to 2015 by Eleanor M. Blyth et al.**

In this paper, the trends in evapotranspiration and its drivers (precipitation and radiation) in Great Britain are investigated. Furthermore, the trend analysis of the different components of evapotranspiration (bare soil, transpiration, interception) was conducted to see which components of the evaporation are contributing to the trend. For this purpose, the authors used the JULES model to have a long-term dataset of evapotranspiration.

As I see, generally, the paper is irregular, and not well-organized. The methodology and the results are not specified clearly. The results are presented like a report, although they are discussed well.

I think the paper provide a little novelty, maybe the good point of the paper is investigating the trend in evaporation components.

Some questions:

1- Page 3, Line 28-31: Authors mentioned that "Assumptions about the processes involved and their interactions (see Wang and Dickinson, 2012 for a review of methods) can have a significant impact on the resulting modelled evaporation (see Schellekens et al, 2017 for overview of model differences).". But they did not say anything about their idea (or the others' idea) for this limitation. I, as a reader, would like to know (after this sentence) that if the authors have any recommendation or any idea for this limitation.

2- Page 5, Line 11 and Fig. 1: I was wondering why the authors presented the figure here, in methodology section? I think it is better to introduce the model firstly, and then present the maps and graphs in the results section.

Some recommendation:

1- The authors used "evapotranspiration" and "evaporation". Please be clear with these two terms. Where "evapotranspiration" should be used and where "evaporation"?

2- The authors should clarify how they calculated the trend. It is a good idea to use the Mann-Kendall test for trend analysis. Furthermore, they can use the modified Mann-Kendall test for the effect of long-term and short-term memory on the trend. The title of the paper shows that the focus of the paper should be on the trend analysis, so one of the most important parts of the methodology should be the trend analysis method.

3- It is not needed to provide the results in both table and figure. Skip one of them and keep only the other one (For example, Table 8 and Fig. 14).

4- I think it is better to combine the results and discussion sections together. Now, there is an inconsistency between the results and discussion which make it ambiguous for the readers to understand the reasons for your results.

5- Page 28, Line 12: Please provide a table in which the differences are shown. It would be easier for the reader to see the differences.

Some typos:

1- Page 2, Line 18: "that" ⟶ "than".

2- Page 3, Line 28: "sunshine and soil moisture, the intercepted of rainfall by plants." ⟶ "sunshine**,** soil moisture **and** the **interception** of rainfall by plants.'

3- Page 4, Line 16: "while" ⟶ "and".

4- Page 5, Line 11: "Figure 1 show maps of the time-average quantities of evapotranspiration" ⟶ "Figure 1 show**s** maps of the time-average quantities of **modelled** evapotranspiration".

5- Page 6, Line 3: "Figure 1. Top and middle rows": There are only 2 rows in the figure, so "**Top and middle rows**" could be skipped.

---

## Referee Comment (RC2) · Anonymous Referee #2 · 13 Jun 2018

**General comments**

The paper analysed the evaporation trend and its drivers in Great Britain using the JULES model. I found the subject and research questions relevant for publication in HESS, and the discussion section is rich. However, I found the research design fundamentally flawed for answering the research questions posed. My main concerns are:

1. **Fixed land cover (and biomass?) over the study period.** The land-use change is fixed in the JULES model, but large scale agricultural abandonment and forest regrowth has occured in the Great Britain during the study period (19.8 million ha agricultural land in 1961, 17 million ha in 2005) (Rounsevell and Reay, 2009). Despite the fundamental role of land in partitioning the precipitation into runoff

and evaporation, the study did not include any sensitivity analyses of the land-use change effect.

2. **Unreferenced key assumptions.** The authors identified interception as the main increasing evaporation component. However, this is based on modelled results and can be biased by for example how interception is modelled and the equation governing the spatial distribution of rainfall intensity (Eq. A1-A3 are all without references, assumptions of the type and spatial distribution of input rainfall on p. 41 are also not referenced or tested). Precipitation data are also subject to large uncertainties. While simulated hydrological fluxes are discussed in comparison to other published results, the trend detected can nevertheless be biased due to model equations and data uncertainties due to e.g., trends in precipitation patterns. The authors appear to be aware of related issues, through e.g., reference to this in the introduction (p. 3 l. 29), but do not discuss or analyse further how different assumptions might interfere with trend analyses. (The authors compare their output with other modelling estimates in Sect 2.3., but focus on evaporation quantities and not trends. Issues of different data use in e.g., GLEAM in different time periods that might compromise its usability for trend analyses are not raised. )

3. **Lack of consideration of alternative explanations.** The authors attribute the change in total evaporation to interception change and precipitation change. However, this is done solely by correlation with radiation and precipitation. It is not motivated why these two have been considered the two main drivers (p. 26 l. 3), despite that many studies in the past have studied the role of e.g., winds (McVicar et al., 2012), land-use change (Sterling et al., 2012), soil moisture limitation (Jung et al., 2010), and $CO_2$ fertilization on evaporation. The effect of $CO_2$ is discussed in Sect 4.6, but could better earn a place in a dedicated section on sensitivity analyses that is cross-referenced to where relevant. Understanding the importance of other factors beyond precipitation and radiation is critically

important for answering research question 3.

4. **Sensitivity and uncertainty analyses could be insightful.** Due to these fundamental issues, rather than trying to answer all three research questions for the entire GB unconvincingly and inadequately, the study could potentially be more insightful if the authors instead tried to answer only one or two of the research questions (even for a more limited region if necessary), but thoroughly. A different research approach could for example have been to use a range of precipitation products, precipitation distribution assumptions, and evaporation modelling approaches to answer question 1-2, rather than simply relying on the CHESS results without testing its sensitivity to a range of different underlying assumptions. Which or what kind of assumptions might overturn the current conclusions?

**Specific comments**

The term "evaporative loss" is useful only in very small systems where the moisture return to other terrestrial areas are of no importance (e.g., loss from a small aquaculture pond), but confusing when used in large natural systems where moisture recycling can occur.

The term "evapotranspiration" could be replaced by total evaporation, see also (Savenije, 2004).

PET could preferably be written as $E_{\mathrm{pot}}$.

Given the central role of trend for the paper, I would suggest the authors to report trend detecting methods and significance more thoroughly.

Sect. 2.2 and 2.3 could go to the Appendix.

Sect 2.4 Can the discussed biases here influence estimates in trends? Perhaps

refer back to this in the discussion section?

P. 21 L. 2 "four" - > "three"

Sect 3.4. Please specify the sample size and p-value of the correlation.

P. 28 Perhaps just briefly remind the reader what type of methods the referenced publications used to arrive at their trend estimates.

The presentation of the results is repetitive and can be further condensed. Please also double-check that all abbreviations and notations are explained.

**References**
Jung, M., Reichstein, M., Ciais, P., Seneviratne, S. I., Sheffield, J., Goulden, M. L., Bonan, G. B., Cescatti, A., Chen, J., de Jeu, R., Dolman, A. J. J., Eugster, W., Gerten, D., Gianelle, D., Gobron, N., Heinke, J., Kimball, J., Law, B. E., Montagnani, L., Mu, Q., Mueller, B., Oleson, K., Papale, D., Richardson, A. D., Roupsard, O., Running, S., Tomelleri, E., Viovy, N., Weber, U., Williams, C., Wood, E., Zaehle, S. and Zhang, K.: Recent decline in the global land evapotranspiration trend due to limited moisture supply, Nature, advance on(7318), 951–4, doi:10.1038/nature09396, 2010.

McVicar, T. R., Roderick, M. L., Donohue, R. J., Li, L. T., Van Niel, T. G., Thomas, A., Grieser, J., Jhajharia, D., Himri, Y., Mahowald, N. M., Mescherskaya, A. V., Kruger, A. C., Rehman, S. and Dinpashoh, Y.: Global review and synthesis of trends in observed terrestrial near-surface wind speeds: Implications for evaporation, J. Hydrol., 416–417, 182–205, doi:10.1016/j.jhydrol.2011.10.024, 2012.

Rounsevell, M. D. A. and Reay, D. S.: Land use and climate change in the UK, Land use policy, 26, S160–S169, doi:10.1016/J.LANDUSEPOL.2009.09.007, 2009.
Savenije, H. H. G.: The importance of interception and why we should delete the term evapotranspiration from our vocabulary, Hydrol. Process., 18(8), 1507–1511, doi:10.1002/hyp.5563, 2004.

Sterling, S. M., Ducharne, A. and Polcher, J.: The impact of global land-cover change on the terrestrial water cycle, Nat. Clim. Chang., 3(4), 385–390, doi:10.1038/nclimate1690, 2012.

---

## Author Comment (AC1) · 11 Jul 2018

Overview. The authors were happy to receive these well considered reviews. Many of the issues can be easily addressed, while others are require more research. I am very much looking forward to working on the issues raised by this paper. We intend this paper to set a benchmark against which such research could be judged. The model (JULES) is used by a wide community of researchers and is the model used in the UK Met Office Unified Model. The configuration of the model used is the standard configurations used by the community for work in the UK. Both reviewers mentioned the issue of the word 'evaporation' verses 'evapotranspiration'. There has been a long debate among micro-meteorologists on the correct term to be used. I was trained to use 'evaporation' for all forms of evaporation (transpiration, interception and soil surface),

but whenever I use that collective term, I find the new generation of environmental scientists object to it. So I have recently started using the term 'evapotranspiration'. For the sake of the majority of readers therefore, I will stick to 'evapotranspiration'. Both reviewers also suggest that the method of how the trend was calculated needs more information. I have included in Section 2.4 a description of this (see A). The textural changes requested by the two reviewers are noted and changes highlighted below. But the more fundamental points are addressed here. Reviewer 2 notes 4 main criticisms. 1. That the paper does not address the land-cover change. This is quite true, although the issue is somewhat exaggerated by the reviewer. The paper by Rounsevell and Reay (2009) is for the whole of the UK (United Kingdom – includes Northern Ireland), not just GB (Great Britain). So the area is not as big as suggested here. The crop model in the current version of JULES is very simple (Osborne et al, 2015, Williams et al, 2017). In terms of evaporation, the crops will be similar to grass. Only half the change from agriculture has been to forest. The total change in forest (also quoted in Rounsevell and Reay 2009) is 1.3m hectares which is less than 5% of the area. This doesn't mean we should study the effect of land cover on the water budget of GB. The issue when we setup the modelling experiment was to see how the climate drivers had altered the water budget. Adding in a land-cover change was likely to complicate the analysis. It would be very interesting to move onto that subject and use this bench-mark paper to quantify the difference. Some text explaining the strategy has been included in the introduction. See B 2. I was particularly excited to read this part of the review. The reviewer is right that the model is indeed sensitive to the assumptions made about rainfall distributions. It is possible that these are changing (with more intense rainfall) and it is important to start to work on this issue. However, what the reviewer is suggesting is a major piece of work in its own right. I hope that this paper demonstrates the importance of such a piece of research. By identifying the role of interception in the evaporation budget and trend, the authors have motivated the kind of study mentioned here. In the discussion section, we have included a discussion of this issue. See C 3. The main drivers were chosen as they were highlighted in the Robinson et al

(2017) paper as being important for evaporation. The wind speeds are indeed reducing (McVicar et al, 2012), but this would give a decrease in evaporation, not an increase. The soil moisture is also dropping globally (Jung et al, 2010) but not in GB, and would also give a decrease in evaporation. The land-use change was not studied here as we wanted to study the role of climate in the modelled evaporation. As discussed above, we are very much looking forward to going into that area. The CO2 fertilisation is a tricky subject mainly because the JULES model is very sensitive to CO2. We studied the outputs of the model, but felt that the uncertainty in the formulation introduced too many issues and that it would distract from the main message of the paper. Again, it is certainly something we would wish to follow up with soon. 4. I think the research approach suggested by the reviewer is a very good one. However, this paper does not describe an experiment. It describes a single configuration. It then asks an important question: for a given standard model in a given standard configuration, what can we learn? What we learn is that there are some aspects of the modelled water cycle (interception) that are more important to the trend than others. This is a unique result and motivates a new priority for hydrologists. We have added some text to the conclusions to highlight this. See D Specifics: Reviewer 1: Page 3, line 28-31. More discussion is given to the issue of limitations of processes assumptions. See E Page 5. I wanted a figure early on to give a flavour of the products being discussed. It is not strictly 'method' then 'results' but I find that approach to be unhelpful sometimes. I like to see the basic product so that I can visualise the rest of the paper. The authors prefer to have both figures (visual comparison for understanding) and a table (quantative, citable). The authors prefer to keep the results and discussions separate. The results are presented as straightforwardly as possible in relation to the original questions. The discussions explore some of the issues raised in more detail. We have added a table of the difference of trend between historic and current analysis. See F Typos all corrected. Reviewer 2: PET is used in the Robinson et al 2017 paper. We would prefer to keep it here for consistency. The biases reported in the results are now mentioned in the discussion of the trends. See G References Osborne, T., Gornall, J., Hooker,

J., Williams, K., Wiltshire, A., Betts, R., and Wheeler, T.: JULES-crop: a parametrisation of crops in the Joint UK Land Environment Simulator, Geosci. Model Dev., 8, 1139-1155, https://doi.org/10.5194/gmd-8-1139-2015, 2015. Williams, K., Gornall, J., Harper, A., Wiltshire, A., Hemming, D., Quaife, T., Arkebauer, T., and Scoby, D.: Evaluation of JULES-crop performance against site observations of irrigated maize from Mead, Nebraska, Geosci. Model Dev., 10, 1291-1320, https://doi.org/10.5194/gmd-10-1291-2017, 2017. A Page 21. Line 12. We calculate the trends as the slope of the linear least squares fit of the annual time series for the different water budget variables. The reported trend errors are calculated as the difference between the trends and the 95 % confidence interval of the linear fit (Robinson et al., 2017). B. Page 4. Line 9. Although the model is complex, there are always simplifications and assumptions made. The model used here is the Joint UK Land Environment Simulator (JULES) which is used by the UK Met Office in the Unified Model. It is also used by a community of land surface and hydrology researchers. The version used in this paper is the new standard configuration used for the UK. The paper is intended to act as a benchmark against which model developments and new research can be compared. In order to interrogate the performance of the model and highlight new research agendas, the following will be addressed: Page 4, line 12. We have not included land-cover change in this analysis. There has been a 5% increase in forest cover over the GB (www.fao.org) which in practise would make a small difference to the water-budgets. But the model analysis for this benchmark paper would be made more complex by this addition variable. Future work would benefit from a comparison of the effect of climate verses land-cover change on the water budgets. C. Page 31. Line 19. The interception model has some uncertainties, and possibly underestimates the effect (see Section 2.2.3). Some of the uncertainty is due to the size of the interception store, some to do with the efficiency of evaporation (the aerodynamic resistance) and some to do with how the rainfall intensity distribution is represented (see Appendix A2) both in time and space. All of these aspects would benefit from further investigation. This paper is only able to highlight the importance of the role of interception in the modelled trend, thus motivating new

research. However, the results of the current model configuration . . . . . . D. Page 33. Line 2. This study set out to explore the long term evolution of the modelled water budget of Great Britain (GB), including how and why it is changing. The model used is the UK community model JULES, used by the Met Office in the Unified Model. The configuration described is the new standard configuration for the UK. E. Page 3. Line 31. For instance: using too shallow roots for the vegetation will result in premature reduction of evaporation in a dry spell (Teuling et al, 2010), ignoring interception and infiltration processes will mean that the surface runoff component will be insensitive to rainfall intensity (Dolman and Gregory, 1992), including water use efficiency processes can alter the long term response of the land to climate change (Prudhomme et al, 2014). F. Page 28. Line 14. Units mm yr-1 yr -1 Precipitation Runoff PET or Evapotranspiration Change in Storage Previous 2.96 1.6 0.7 or 0.77 0.5 New 2.96 2.16 0.87 0 Table 9. Overview of previous and new estimates of trends in the water budget for GB. G Page 28. Line 28. It should be noted that the trends calculated form these model results are subject to the model biases which are shown in Section 2. The overall evaporation is low compared to the observed and the interception is particularly low compared to the estimates from the Forestry Commission (Nisbet, 2005). If it is true that the trend in evapotranspiration is higher than the trend in PET due to the presence of interception, then it is possible that the true trend might be higher still if that is being under-estimated. References: Dolman, H. and Gregory, D., 1992. The parameterisation of rainfall interception in GCMs. Quarterly J. of Roy. Met. Soc., 118, 455-467 Teuling, A.J., Seneviratne, S.I., Stockli, R., Reichstein, M., Moors, E., Ciais, P., Luyssaert, S., van den Hurk, Amman, C., Bernhofer, C., Dellwik, E., Gianelle, D., Gielen, B., Grunwalkd, T., Klumpp, K., Montagnani, L., Moureaux, C., Sottocornala, M and Wohlfahrt, G. 2010. Contrasting response of European forest and grassland energy exchange to heatwaves. Nature Geoscience, 3, pages722–727